# How does China's green factory policy affect substantive green innovation?

Xiangshu Dong[1], Yongjiao Du[1], Xiang Xiao[2]*

**1** School of Economics, Capital University of Economics and Business, Beijing, China, **2** School of Marxism, Central University of Finance and Economics, Beijing, China

* ganyu8888@126.com

## Abstract

Green factory evaluation (GFE) is a green industry policy in China with the goal of accelerating green manufacturing. How does GFE influence corporate substantive green innovation (SUGI)? Existing research has yet to provide definitive results. Based on GFE policy, this paper utilizes the data of China's A-share listed manufacturing firms spanning 2012–2021 by the time-varying difference-in-differences model to investigate the impact of GFE on SUGI. The results indicate that GFE can significantly improve corporate SUGI. The mechanism results suggest that GFE promotes corporate SUGI by limiting greenwashing and enhancing industry-university-research cooperation. Additionally, we find the impact of GFE has greatly boosted corporate SUGI in industries characterized by high energy consumption and pollution levels, suggesting a potential disruption of technological path dependencies within these sectors. Moreover, firms with extensive external information disclosure exhibit stronger enhancements in SUGI under GFE. The conclusions confirm the effect and internal logic of GFE's influence on corporate SUGI and provide a valuable advice for relevant policies aimed at improving green development.

## 1. Introduction

In recent years, China's industrialization has experienced sustained and rapid growth. While this has driven significant economic development and established a comprehensive manufacturing system, it has also resulted in increasingly severe environmental pollution issues [1–3]. As a key driver in promoting green development, substantive green innovation (SUGI) provides effective support for improving environmental quality and achieving sustainable development goals. Research on SUGI is currently a key focus in academia [4–7].

However, businesses often lack the resources and incentives to promote SUGI. On the one hand, compared to traditional green innovation, SUGI involves high risk, high investment, high uncertainty, and long cycles [8], making it harder to gain support within enterprises. On the other hand, in the absence of a strict environmental policy and

**Data availability statement:** All relevant data are within the manuscript and available at the public database Figshare and obtained the DOI link https://doi.org/10.6084/m9.figshare.28646624.

**Funding:** This study was funded by the R&D Program of Beijing Municipal Education Commission: Research on Beijing-Tianjin-Hebei Cooperative Development Strategy and Total Factor Productivity in Manufacturing Industry (No:SM202110038014); Science and Technology Innovation Programme for Academic Degree Graduate Students of Capital University of Economics and Business (2024KJCX030).

**Competing interests:** The authors have declared that no competing interests exist.

regulatory framework, most enterprises tend to overemphasize the 'quantity' of green innovations to achieve superficial environmental performance [9–11], while ignoring SUGI that genuinely contribute to the long-term sustainable development of enterprises. As a form of industrial policy aimed at environmental goals [12], green industrial policy can effectively overcome the limitations of traditional industrial and environmental policies in promoting green development. It serves as an effective approach to mitigating 'market failure' and stimulating the vitality of green innovation within enterprises [13–18]. However, existing literature has paid limited attention to how green industrial policy influences SUGI. In this context, exploring whether green industrial policy can promote corporate SUGI is highly important for accelerating the green transformation of the manufacturing industry and achieving sustainable development.

Unlike traditional environmental regulation, which focuses on end-of-pipe treatment and penalties, green industrial policy aims to guide enterprises in enhancing the research, development, and application of cleaner production technologies, as well as in developing green processes and products. One of its core goals is to promote the green transformation of enterprises. As an important component of China's green industrial policy, the green factory evaluation (GFE), officially launched in 2016, provides a valuable opportunity to explore how green industrial policy influences corporate SUGI. In fact, some scholars have already highlighted the significant role of macro-level pilot policies [19], such as the GFE, in influencing firms' production behavior. They indicated that the GFE, as a green industrial policy, not only significantly enhances labor productivity within firms [20] but also contributes to promoting the green development of firms [21]. However, these studies often focus on the economic and social impacts of the GFE or its effectiveness in promoting firms' overall green innovation, while the systematic study of its impact on SUGI is still limited. As the core of a green manufacturing system, does the GFE promote corporate SUGI, and what are the mechanisms through which it exerts its impact? Answering this question not only helps clarify the relationship between the GFE and SUGI and offers new perspectives for exploring the theoretical study of how green industrial policy influences SUGI, but also provides more targeted practical insights for promoting the green transformation of the manufacturing industry.

Based on this, this paper seeks to systematically investigate the impact and internal logic of GFE on corporate SUGI. Specifically, based on GFE policy, we employ the time-varying difference-in-differences (TDID) model to empirically examine the link between GFE and corporate SUGI. The results show that GFE has a positive impact on promoting corporate SUGI, and the influencing mechanisms are greenwashing and industry-university-research cooperation (IURO). Further, this paper also finds that GFE significantly contributes to firms' SUGI in industries with high energy consumption and pollution levels, as well as in firms characterized by higher levels of external information disclosure. The research conclusions provide essential evidence supporting GFE's role in promoting the green development of micro-enterprises. Our study makes three important contributions to the existing literature:

First, this study focuses on the substantive performance of green innovation by integrating the GFE and corporate SUGI into the same research framework, offering

a new perspective for exploring the drivers of SUGI. Existing studies on GFE tend to focus on exploring its impact on labor productivity or green innovation [20–21], largely ignoring its role in promoting SUGI, a high-quality form of green innovation. However, this study further explores how the GFE, as a green industrial policy, promotes corporate SUGI. Our research not only contributes to the literature on the effectiveness of GFE and adds to the ongoing discussion about the efficacy of green industrial policies, but also provides new empirical evidence on how emerging manufacturing countries can effectively stimulate SUGI in the manufacturing sector.

Second, based on the characteristics of SUGI, our study conducts a comprehensive analysis of the intricate relationship between GFE and SUGI, and finds that greenwashing and IURO are the mechanisms through which GFE affects SUGI. By focusing on the information asymmetry and the resource-based theory, we demonstrate that GFE effectively addresses and alleviates the challenges of strategic green innovation and resource constraints faced by firms during the green innovation process, thereby promoting SUGI. Unlike existing studies that explore the impact mechanism of green industrial policy on SUGI from the perspectives of environmental investment and human capital investment [22], this study further clarifies the intrinsic mechanism through which GFE promotes SUGI. These findings help unveil the 'black box' of GFE's influence on SUGI through the lens of corporate behavior, thereby providing an internal logic for various market participants to explore how green industrial policies can help firms achieve SUGI goals.

Third, our study also examines how the GFE affects firms' SUGI differently in various contexts, providing valuable empirical evidence for understanding the role of green industrial policies in guiding firms' green innovation. Existing studies have not yet reached a consensus on how green industrial policies affect corporate green development [15,23,24]. By focusing on GFE as a green industrial policy, we find that GFE promotes SUGI in the heavy chemical industry. This provides valuable insights for heavy chemical enterprises to break path dependence and accelerate their green transformation. Additionally, we find that GFE significantly enhances SUGI when firms are subject to stronger external information disclosure. These findings provide clearer policy insights into how to effectively implement green industrial policies and facilitate the green transformation of the manufacturing industry in the current context.

The study is organized as follows: Section 2 provides literature review. Section 3 covers theoretical analysis and hypotheses. Section 4 details variables, data, and the empirical models. Section 5 presents the empirical analysis and results. Section 6 offers conclusions and policy implications.

## 2. Literature review

GFE, being a significant component of industrial orientation strategies, presents an inherent and rational occasion to advance the environmentally sustainable progress of the manufacturing sector. Nevertheless, scholarly literature regarding the investigation of GFE is scarce. Moreover, the precise nature of the connection between GFE and SUGI is uncertain. Thankfully, an abundance of scholarly research belonging to SUGI and green industrial policy can furnish us with a robust theoretical foundation.

Academic circles have been preoccupied with green innovation for quite some time [25–27]. The overlapping nature of the positive externalities of knowledge spillover and negative externalities of pollution resulting from the dual externalities of green innovation may diminish the private benefits of green innovation relative to its social benefits [28], thereby demotivating businesses to pursue green innovation. To encourage green innovation in this context, governments typically consider implementing punitive or positive incentives [29–30]. The relationship between green innovation and environmental policy has not yet been conclusively established by the research that currently exists. From the perspectives of low carbon city policy [31–33], green public procurement [34], green financial policy [14,35], green credit policy [36], and environmental policy mechanisms [37], a considerable number of scholars have examined the relationship between green policy and green innovation. They advocate for the notion that green policies have the potential to stimulate green innovation and enterprise green development. Conversely, due to the significant information asymmetry that exists in the realm of green innovation, a number of studies have confirmed that environmental policy can significantly enhance corporate

environmental performance, but this may lead to a significant increase in strategic innovation [38]. When faced with such a situation, one might contemplate the impact of GFE on the selection of environmentally friendly innovations and whether it will serve as the primary impetus for SUGI in the manufacturing sector [39]. This subject continues to be amenable to further investigation and discussion.

GFE is an integral component of the green industry policy in China [40]. In contrast to traditional industry policies, the green industry policy encompasses environmental and industrial concerns simultaneously [41]. Assessing the effectiveness of green industry policies is a critical component of environmental economics and a crucial policy instrument for nations seeking to achieve green development [17–18]. The GFE, which was implemented in 2016, is an essential component of the industrial orientation policy and a crucial policy in the advancement of green manufacturing in China. In recent years, there has been a gradual increase in the scholarly focus on GFE. Chen [40] conducted an in-depth analysis of the interconnection between capital markets and green industrial policies, using GFE as their research subject. Zhu [42] verified that enterprises' green innovation levels have considerably increased since receiving green factory designations, this is advantageous for achieving a breakthrough in the quality of innovation as opposed to its quantity. These aforementioned studies tend to explain the effect of green innovation at the aggregate level, with little emphasis on the various green innovation options [43]. The research mentioned above provides significant insights that can enhance our study. However, it also raises certain concerns that merit further examination. To what extent does GFE will impact SUGI? Additional research is warranted at this time.

## 3. Theoretical analysis and research hypothesis

### 3.1. GFE and SUGI

GFE is a crucial policy to accomplish green manufacturing for China. Its core purpose is to promote green manufacturing by establishing demonstration benchmarks in the industry and guiding and standardizing the factory. In an effort to establish a green manufacturing system, Made in China 2025 had previously proposed the notion of a green factory. The Ministry of Industry and Information Technology (MIIT), the National Development and Reform Commission, and the Ministry of Finance collaborated in 2016 to publish the Implementation Guide of the Green Manufacturing Project (2016–2020). This guide outlined the explicit intention to conduct green evaluations and marked the official introduction of the GFE. The three levels of GFE are clearly delineated as follows: national, provincial, and municipal. The MIIT painstakingly supervises the entire application and selection process for national green factories in order to identify GFE. Subsequently, departments of industry and information technology at all levels present the list of applications step by step, and lastly, MIIT reviews and verifies the GFE enterprises. It is important to note that GFE covers four aspects: green evaluation standard formulation, third-party evaluation and certification, policy incentives, and continual supervision and evaluation.

First, the General Rules for the Evaluation of Green Factory (GREGF), based on product life cycle management, outline a comprehensive evaluation system for green factories. These indicators include general requirements, infrastructure, management system, energy and resource input, products, and environmental emissions. These criteria are developed and published as national standards, considerably decreasing the black box issue created by opaque evaluation benchmarks. Second, in terms of third-party review, if a company's indicators, such as pollutant discharge and energy usage, exceed the industry average, the enterprise can request reports from third-party intermediaries on its own. Moreover, the assessment must be chosen and acknowledged across multiple tiers, including endorsement by provincial industrial and information departments, demonstration by pertinent specialists, review, and online publicity. Ultimately, this concludes with the determination of the list of national green factories within the region. Furthermore, with regard to policy incentives, businesses assessed as green factories may qualify for one-time rewards, special fund assistance, preferential loan interest rates, special credit lines, and additional government resources. To illustrate, green factory enterprises at the municipal and national levels are granted 300,000 and 600,000 yuan, respectively, in Tianjin. The corresponding institutional guarantee for the assessed enterprises' fund utilization is provided by GFE as a fundamental element of green industrial policy. Moreover, with regard to ongoing oversight, GFE dynamically manages the accredited businesses in accordance

with the 'review every three years' principle. Neglecting to adhere to the prescribed benchmarks with regard to environmental performance metrics will result in the enterprise's removal from the GFE list and a subsequent three-year ineligibility to reapply. Implementing mandatory post-event monitoring is beneficial in ensuring that the GFE enterprise optimizes the demonstrated effect of 'location-specific efforts' and enhances their environmental performance. GFE, which is considered a novel form of green industrial policy, has implemented a framework for transparent evaluation, provided enterprises with financial assistance and green certification, and prioritized dynamic oversight and third-party evaluation. GFE is advantageous for mitigating the potential incentive distortion that may arise from conventional green policies, thereby facilitate SUGI. Based on this, the research hypothesis is stated as follows:

**H1:** GFE can positively promote corporate SUGI.

### 3.2. GFE, greenwashing and SUGI

GFE can reduce the information asymmetry between government policies and businesses, thereby preventing businesses from engaging in greenwashing and increasing SUGI. In a broad sense, greenwashing generally refers to the practice of companies engaging in superficial green initiatives without actually implementing practical measures to address environmental issues. An essential incentive for businesses to partake in greenwashing practices is to secure government preferential policies through the presentation of a favorable green image [10]. Greenwashing has the potential to relegate enterprise R&D to symbolic strategic innovation, where substantive progress is merely nominal. This procedure will negatively affect SUGI, which ordinarily necessitates greater risk, greater investment, and more complex green technology. Nonetheless, enterprise greenwashing may be mitigated through the application of GFE.

First, in the process of policy design, GFE restricts enterprises' strategic green behaviors through the implementation of preventative measures in advance and continuous oversight throughout the process. One significant benefit of disclosing prior standards is that it significantly reduces the likelihood of business fraud. In particular, the publication of GREGF has done its utmost to eliminate obscure criteria from enterprise green award evaluations. Moreover, the implementation of GFE mitigates the effects of government failure to a certain degree, enhances multi-agent oversight, and raises the bar for collusion. In addition, self-evaluation, evaluation by third-party organizations, recommendation from provincial industrial and information departments, expert demonstration and review by the MIIT, and online publicity are all necessary for an organization to be recognized as a green factory. By establishing a chain of supervision that includes professional institutions, local authorities, and the central government, oversight vulnerabilities that could potentially result from relying exclusively on individual inspection points are effectively avoided.

Second, GFE emphasizes the establishment of a mechanism for long-term restraint. In contrast to other green honors that operate on a short-term basis, the GFE incorporates a subsequent longer-term constraint mechanism. In addition to periodic spot checks and reviews—under which the evaluated enterprises are subject to dynamic management based on the principle of 'review every three years'—and the removal from the existing list of enterprises that do not meet the review standards—the constraint is also reflected in a series of mandatory information disclosure requirements for evaluated enterprises. According to [40], the implementation of long-term monitoring of evaluation outcomes for green factories will serve to fortify peer oversight, mitigate information asymmetry, substantially diminish moral hazard subsequent to the evaluation process, and incentivize the ongoing enhancement of environmental performance. On the basis of the preceding analysis, we put forward the following hypotheses:

**H2**: GFE can promote SUGI by alleviating greenwashing.

### 3.3. GFE, IURO and SUGI

The implementation of the GFE serves as a catalyst for the advancement of IURO, benefiting the manufacturing industry's SUGI. Green innovation encompasses a diverse set of knowledge fields. Green technology has become increasingly

complex in recent years, owing to the rapid acceleration of technological innovation. In this context, enterprises face significant challenges in mastering all technologies on their own. However, Chinese universities and research institutes have strong capabilities in the field of green scientific research. GFE can promote IURO by enhancing SUGI.

The resource-based theory suggests that enterprises with more resources are more likely to engage in innovative behaviors. Enterprises that are successfully elected as green factories can benefit not only from government subsidies, but also from other preferential policies implemented by local governments, such as tax breaks, government procurement, priority administrative examination and approval, and exemption from peak shifting production. At the same time, this policy will send out a green signal, allowing businesses to gain the favor of external investors and support for green credit. This process promotes green innovation and accelerates enterprise green transformation [44]. In this context, the implementation of GFE policy will help to strengthen green technology cooperation between enterprises with more resources and other subjects, thereby providing a good platform and conditions for the breakthrough of SUGI.

Besides, the implementation of GFE reduces information asymmetry among enterprises, universities, and research institutes. The nomination of enterprises for green factory status is equivalent to obtaining national green certification due to the fact that GFE is subject to self-evaluation and third-party evaluation, provincial selection and recommendation, and confirmation by the MIIT. This will elicit a favorable, affirmative response from academic and research institutions, universities, and other organizations, thereby enabling a partial reduction in the expenses associated with communication between businesses and these institutions. Conversely, universities and research institutes are also incentivized to collaborate with businesses deemed to be green factories in order to conduct green innovation activities collectively and secure additional government funding. A procedure analogous to the one described above will aid businesses in conducting IURO activities and promoting SUGI. Based on this, we put forth the following hypothesis:

**H3**: GFE can promote SUGI by enhancing IURO.

## 4. Research design

### 4.1. Variable selection

#### 4.1.1. Green factory evaluation (GFE).
GFE is the explanatory variable. If firm $i$ is selected as an GFE firm in year $t$, it is classified as a treatment group with GFE equals 1; otherwise, it is classified as a control group with GFE equals 0. Since GFE involves multiple periods, we choose the TDID model to explore the impact of GFE on corporate SUGI. From 2017 to 2020, the MIIT identified five batches of green factory lists and evaluated them at different times, as stated in the General Office of the MIIT's Notice on Developing Green Manufacturing System and related documents. Specifically, the details of the enterprises assessed as green factories are as follows: 201 in 2017, and in 2018, GFE was divided into two batches: 208 and 391. In 2019, it was 611, and in 2020, it was 724. To ensure the reliability of enterprise sample data, we use manual screening to compare GFE with published information from Cninfo and Qichacha search databases. After removing the sample of enterprises with anomalies and mismatches, we had 351 listed companies in green factories, for a total sample size of 1059.

The distribution of GFE provides a better understanding of its characteristics. First, to clearly characterize the temporal evolution of GFE, we plot its time trend. Fig 1 shows the changes in the number of companies participating in GFE between 2012 and 2021. It can be observed that the number of green factories has exhibited a year-by-year increasing trend since the release of the first list of green factories. This demonstrates that the implementation of GFE has yielded positive outcomes. Its successful practice not only encourages more enterprises to actively engage in green manufacturing through the demonstration effect but also strengthens the government's confidence and commitment to further expand the scope of the pilot program.

Second, to portray the distribution of GFE across different manufacturing industries in a more detailed manner, we mapped the industry distribution of GFE based on the Guidelines for Industry Classification of Listed Companies (revised

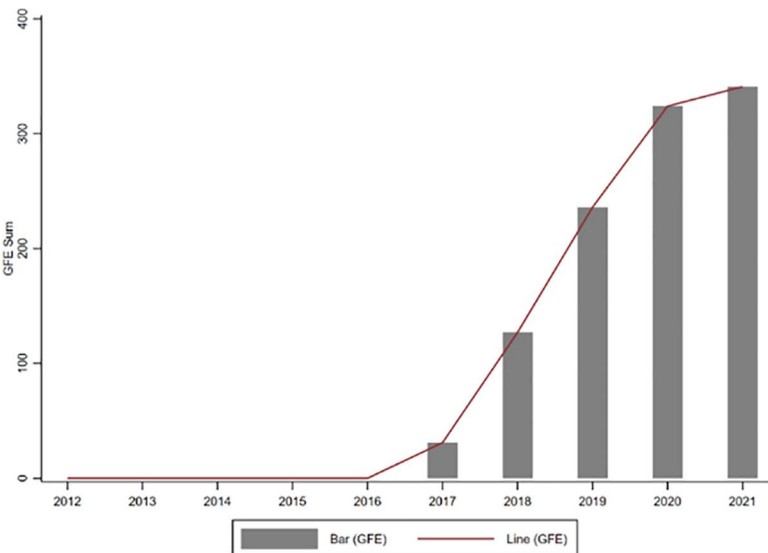

**Fig 1. Time trend of GFE sum.**

in 2012). Specifically, C13 = Processing of Agricultural and Sideline Products, C14 = Food Manufacturing, C15 = Manufacturing of Alcohol, Beverage and Refined Tea, C17 = Textile Industry, C18 = Textile and Garment Industry, C19 = Leather, Fur, Feather and their Products and Footwear Industry, C20 = Wood Processing and Manufacturing of Wood, Bamboo, Rattan, Palm, and Straw Products, C21 = Furniture Manufacturing, C22 = Papermaking and Paper Products Industry, C23 = Printing and Record Medium Reproduction Industry, C24 = Manufacturing of Cultural, Educational, Artistic, Sporting, and Entertainment Goods, C25 = Petroleum Processing, Coking, and Nuclear Fuel Processing Industry, C26 = Chemical Raw Materials and Chemical Products Manufacturing, C27 = Pharmaceutical Manufacturing, C28 = Chemical Fiber Manufacturing, C29 = Rubber and Plastic Products Industry, C30 = Non-Metallic Mineral Products Industry, C31 = Ferrous Metal Smelting and Rolling Processing Industry, C32 = Non-Ferrous Metal Smelting and Rolling Processing Industry, C33 = Metal Products Industry, C34 = General Equipment Manufacturing, C35 = Special Equipment Manufacturing, C36 = Automobile Manufacturing, C37 = Railway, Ship, Aerospace and Other Transportation Equipment Manufacturing, C38 = Electrical Machinery and Equipment Manufacturing, C39 = Manufacturing of Computers, Communications, and Other Electronic Equipment, C40 = Manufacturing of Instruments and Meters, C41 = Other Manufacturing, C42 = Comprehensive Utilization of Waste Resources Industry. Fig 2 shows the number of firms that have been selected for GFE programs across different manufacturing sectors. It demonstrates that a higher number of companies have been approved as green factories in the following sectors: pharmaceutical manufacturing (C27), electrical machinery and equipment manufacturing (C38), manufacturing of computers, communications, and other electronic equipment (C39), chemical raw materials and chemical products manufacturing (C26), and automobile manufacturing (C36). As key industries in the transformation and upgrading of the manufacturing sector, most of these industries are characterized by high pollution and high emissions. The GFE enables these industries to ensure sustainable development while actively pursuing green transformation, further exerting a 'point-to-point' demonstration effect.

**4.1.2. Substantive green innovation (SUGI).** Green patents are an important indicator of the output of green innovation knowledge outcomes for enterprises and are widely used to measure corporate green innovation [45–46]. They primarily include two types: green invention patents and green utility model patents. Although both types of green patents can be used to measure corporate green innovation, they differ significantly in terms of technical content and the difficulty of filing.

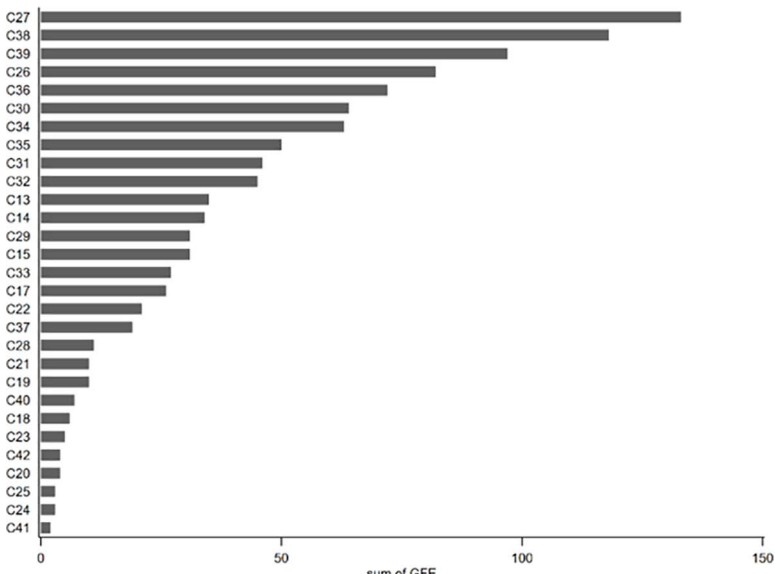

**Fig 2. Distribution of GFE sum in manufacturing industry.**

According to the China's Patent Law, invention patents typically refer to a new technical solution related to a product, a method, or an improvement thereof, whereas utility model patents refer to new technical solutions pertaining to the shape, structure, or combination of products that are suitable for practical application. In terms of examination content, the examination process for invention patents is more stringent. It includes not only a formal examination but also the requirement that the patent demonstrates 'outstanding substantive features and significant progress' [7,30]. This necessitates a higher degree of technical difficulty and quality. The examination process for utility model patents is relatively simpler, focusing primarily on the form and practical applicability of their technical solutions, and requiring a lower level of technical content. In terms of examination duration, the examination cycle for invention patents is longer, typically taking 2–3 years or even more, and includes both preliminary examination and substantive examination. In contrast, the examination period for utility model patents is shorter, typically lasting 6–12 months, and involves only a preliminary examination. Based on the above definition, the technical content required for green invention patents are significantly higher than that required for green utility model patents.

The measurement of SUGI is a prominent research topic in the field of green innovation and serves as the core variable of this study. However, there is currently no unified or standardized method for measuring SUGI within the academia. Many scholars use the number of green invention patent applications to measure SUGI [6,7,39,47]. Yet, a few scholars employ the total number of green patent applications to measure SUGI [48]. Since the total number of green patent applications reflects a company's overall level of green innovation, it does not distinguish between specific types of patents. This may result in an overestimation of an enterprise's SUGI. Compared to green utility model patents, green invention patents are more groundbreaking and creative in nature, higher in quality, and better at reflecting SUGI. Therefore, in line with the definition of patent law and the research of most scholars, we use the number of green invention patent applications to measure corporate SUGI. This approach has been widely recognized in academia.

Specifically, we identify green patents based on the indexed list of environmentally friendly International Patent Classification (IPC) published by the World Intellectual Property Organization (WIPO) in 2010. The rationale for selecting this classification framework is as follows: First, our study period (2012–2021) closely aligns with the temporal coverage of the 2010 WIPO classification. This version captures approximately 90% of green patents within our sample period, demonstrating

strong representativeness. Second, as the first systematic green patent classification framework developed by WIPO, the 2010 IPC list encompasses approximately 200 environment-related technology categories, supported by comprehensive patent information. Its widespread adoption by global patent offices and academics from its complete IPC mapping and long-term stability. For instance, scholars such as [49–52] have applied this classification framework in their studies. Using this classification system for green patent identification helps ensure both the reliability of research findings and their international comparability. The list categorizes green technologies into seven categories: alternative energy production, transportation, energy conservation, waste management, agriculture and forestry, administrative, regulatory, or design aspects, and nuclear power generation. Following the above definition criteria, this study calculates the number of green invention patents through the following steps. First, we download information on patent applications and IPC classification numbers of listed Chinese manufacturing firms from the incoPat database. Second, based on the IPC classification numbers of green patents in the 'International Patent Green Classification List,' we manually search and match the IPC classification numbers of enterprises in the incoPat database with green patent information across the seven categories, thereby screening out the green patents. Third, we count and summarize the number of corporate green invention patent applications each year and used it as the measure of SUGI. To timely assess the impact of GFE on SUGI, following [53], we measure SUGI using the number of green invention patent applications with a one-period lag. Since many patents exhibit a zero and right-skewed distribution, we add one to the number of green invention patents and take the logarithm.

Notably, some studies suggest that compared to the quantity of green patent applications or grants, the citation status of corporate green patents can reflect the innovation value and quality of green technologies [54]. To ensure a more comprehensive and reliable measurement of SUGI, we applied quality-based indicators such as citation counts and knowledge breadth of green invention patents as alternative variables for SUGI. Additionally, it is recognized that enterprises may use green management techniques and methods to improve their production processes, thereby enhancing their green innovation capabilities. Such innovations may not be captured by the number of patents. Therefore, following Zhao [55], we also calculated green management innovation and used it as an alternative proxy for SUGI. In the robustness tests, we present the estimation results using the above two indicators as the explained variables.

To gain a deeper understanding of the characteristics of SUGI in manufacturing firms, we plotted the time trend and industry distribution of SUGI separately. Fig 3 shows the trend of SUGI over time. The results indicate that the overall SUGI among manufacturing firms exhibited a gradual upward trend during the sample period. However, it is worth noting that this upward trend reversed in 2018, showing a brief decline. SUGI represents a high-quality innovation behavior that requires significant R&D investment and advanced technology. Due to the shocks and uncertainties in the global economic situation, firms likely faced increased cost pressures during this period, which reduced their willingness to invest in SUGI. After 2020, SUGI resumed a gradual upward trend. With the carbon neutrality goal, enterprises are increasingly recognizing the importance of SUGI for their sustainable development and have begun to consistently strengthen their investments in cutting-edge technology and R&D. This is likely to drive an increase in SUGI activities among firms.

Fig 4 shows the distribution of SUGI across different manufacturing industries. The results indicate that the top five industries with the highest SUGI rankings are the manufacturing of computers, communications, and other electronic equipment (C39), electrical machinery and equipment manufacturing (C38), chemical raw materials and chemical products manufacturing (C26), automobile manufacturing (C36), and special equipment manufacturing (C35). Most of these industries are technology-intensive, characterized by higher R&D investment intensity and stronger technological innovation capabilities. They also perform better in the R&D and transformation of SUGI.

**4.1.3. Control variables.** In accordance with [7,56], we select a series of control variables that may influence the SUGI of enterprises to mitigate the influence of other potential factors on the research conclusions. These variables include company age (AGE), company size (SIZE), management fee rate (MFR), financial leverage ratio (FLR), the shareholding ratio of the largest shareholder (TOP1), growth opportunity (GO), proportion of fixed assets (PFA), profitability (PRO), and cash flow situation (CFS). Variables' descriptions are summarized in Table 1.

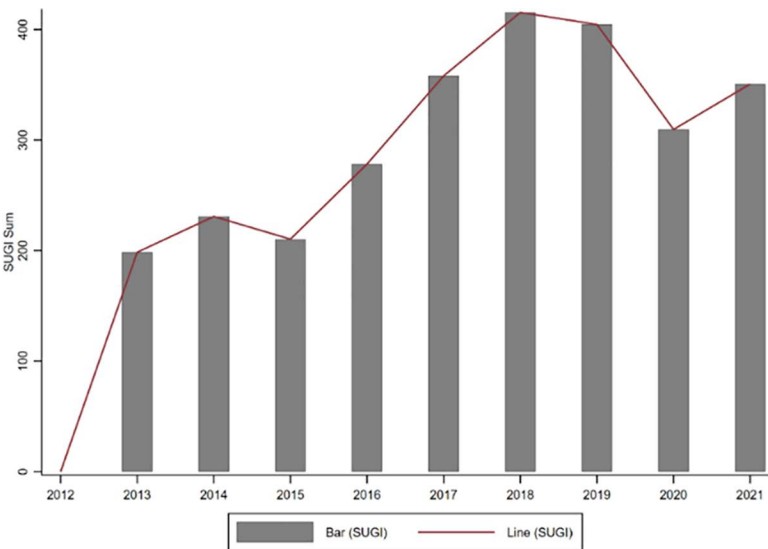

**Fig 3. Time trend of SUGI sum.**

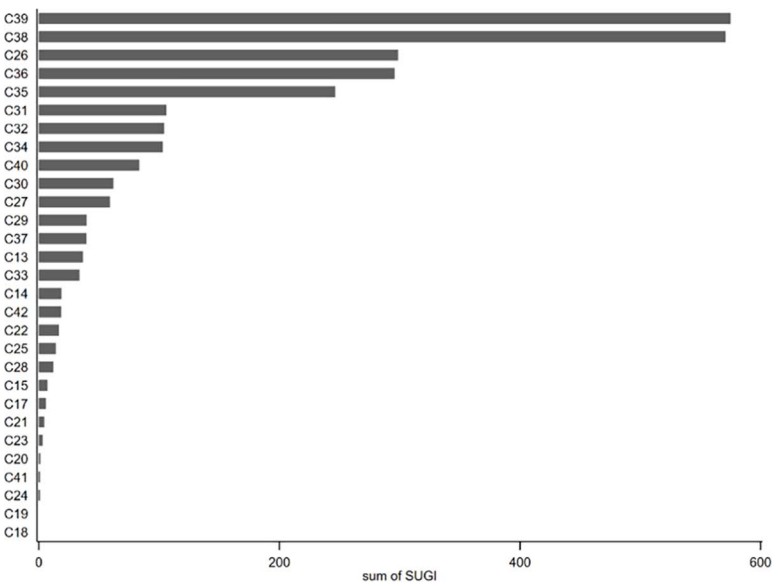

**Fig 4. Distribution of SUGI sum in manufacturing industry.**

## 4.2. Data and sample

Considering the GFE was implemented in 2017, on the basis of available data, we select all Chinese A-share manufacturing companies listed on the Shanghai and Shenzhen Stock Exchanges between 2012 and 2021 as an initial sample, and utilize TDID to examine the impact of GFE on SUGI. Following that, the sample is processed in accordance with the subsequent criteria: (1) Eliminating the financial category and the companies, including ST, ST*, and PT, whose financial status was abnormal that year. (2) To avoid the error of data information disclosure, we have excluded the firms that went

**Table 1. The variable description.**

| Var | Name | Definition |
|---|---|---|
| GFE | Green factory evaluation | If a company is evaluated as green factory, the GFE is 1, otherwise, it is 0 |
| SUGI | Substantive green innovation | The value of the logarithm of the number of green invention patents increased by one, then lagged by one period |
| AGE | Company age | The logarithm of the difference between the observation year and the IPO year of the enterprise. |
| SIZE | Company size | The logarithm of total assets |
| MFR | Management fee rate | The ratio of total management expenses to total assets |
| FLR | Financial leverage ratio | The ratio of total assets to total liabilities |
| TOP1 | The shareholding ratio of the largest shareholder | The ratio of the number of shares held by the largest shareholder to the total share capital of the enterprise |
| GO | Growth opportunity | The ratio of market price (share price) to replacement cost |
| PFA | Proportion of fixed assets | The ratio of net fixed assets to total assets |
| PRO | Profitability | The ratio of profit to total assets |
| CFS | Cash flow situation | The ratio of net cash flow from operating activities to total assets |

public during that year. (3) The process of identifying and eliminating businesses that have a quantity of missing values in the primary variables. (4) To reduce the impact of outliers on our research, all continuous variables (excluding dummy variables) are winsorized at the 1% level in each tail. Among the data sources utilized for this study are: patent data at the enterprise level is obtained from the incoPat patent database. Other control variables are obtained from the CNRDS and CSMAR databases.

### 4.3. Model construction

The DID model is widely used in policy evaluation research due to its strengths in causal identification. The model can effectively control for the interference of other policies while eliminating ex ante differences between the treatment group (pilot firms) and the control group (non-pilot firms), thereby accurately estimating the net effect of policy shocks on firms' behavior. Given that the list of GFE firms is enacted in batches, firms in the treatment group are subject to the policy at different points in time, resulting in variations in treatment timing. The TDID is suitable for studying the effects of policies implemented at different times on different individuals. It can address issues such as inconsistent treatment timing and potential estimation bias arising from traditional DID, thereby providing a more efficient estimator. Based on this, we employ the TDID model to examine the impact of GFE on firms' SUGI. The specific model settings are as follows:

$$SUGI_{it} = \beta_0 + \beta_1 GFE_{it} + \beta_2 X_{it} + \theta_i + \delta_t + \varepsilon_{it} \qquad (1)$$

$SUGI_{it}$ is the SUGI by enterprise $i$ in year $t$. $GFE_{it}$ is a dummy variable that equals 1 if enterprise $i$ is evaluated as green factory in year t, otherwise, it is 0. $X_{it}$ represents the set of control variables. $\theta_i$ is the firm fixed effect, $\delta_t$ is the year fixed effect, $\varepsilon_{it}$ is a random error term.

## 5. Empirical result

### 5.1. Descriptive statistics

The descriptive statistics of the primary variables employed in our study are displayed in Table 2. These findings indicate that the average GFE for the sample of manufacturing enterprises spanning the years 2012–2021 is 0.067, accompanied by a standard deviation of 0.250. It suggests that coverage of the GFE among Chinese businesses is relatively limited, that a substantial number of businesses have not yet participated in the process of GFE, and that this has led to

Table 2. Descriptive statistics of main variables.

| Var | Obs | Mean | SD | Min | Max |
|------|------|------|------|------|------|
| SUGI | 14637 | 0.188 | 0.537 | 0 | 5.147 |
| GFE | 15851 | 0.067 | 0.250 | 0 | 1 |
| AGE | 15851 | 2.075 | 0.764 | 0.693 | 3.296 |
| SIZE | 15851 | 22.149 | 1.177 | 19.124 | 26.981 |
| MFR | 15851 | 0.088 | 0.077 | 0.007 | 1.070 |
| FLR | 15851 | 0.408 | 0.191 | 0.066 | 0.908 |
| TOP1 | 15851 | 33.254 | 13.816 | 9.23 | 74.66 |
| GO | 15851 | 2.169 | 1.368 | 0.866 | 8.371 |
| PFA | 15851 | 0.230 | 0.137 | 0.0021 | 0.722 |
| PRO | 15851 | 0.036 | 0.065 | −0.265 | 0.193 |
| CFS | 15851 | 0.056 | 0.078 | −0.216 | 0.310 |

enormous disparities in the degree of GFE implementation among sample businesses. The calculated means and standard deviations for SUGI is 0.188. It suggests that the level of SUGI in Chinese enterprises is still in its early stages, and discernible discrepancies do indeed exist. AGE, SIZE, MFR, FLR, TOP1, GO, PFA, PRO, and CFS have the following respective means: 2.075, 22.149, 0.088, 0.408, 33.254, 2.169, 0.230, 0.036, and 0.056. These statistics are consistent with prior research discussed in the literature.

## 5.2. Benchmark regression

Table 3 presents the empirical results of the benchmark Model (1). Column (1) presents the regression coefficient of GFE on SUGI, which demonstrates a positive significance at the 1% level without the firm and year effects. In accordance with Columns (1), Column (2) adds controls variables, and the regression coefficient for GFE is 0.076, which is significantly positive at the 1% level, showing that GFE can significantly improve the SUGI of manufacturing enterprises. H1 is confirmed. In fact, compared with green utility model patents, green invention patents are more groundbreaking and creative [6,7,47]. They can effectively promote technological progress and reduce pollution emissions, serving as the real driving force to enhance the vitality of green innovation in the manufacturing industry. In the context of vigorously promoting green manufacturing, firms with GFE have strong incentives to enhance their corporate green innovation activities [21] to accelerate the realization of green transformation, particularly in SUGI focused on green invention patents.

## 5.3. Robustness test

**5.3.1. Parallel trend test.** Generally speaking, if corporate SUGI is affected by unpredictable factors, enterprises implemented GFE policy will not exhibit time-based differences in their SUGI. Moreover, if low SUGI itself drives the implementation of GFE, then before GFE is implemented, the SUGI of some enterprises would differ from those that have not implemented GFE. Based on this, this study employs the parallel trend test to examine the dynamic effect of GFE on firm's SUGI. Fig 5 displays the results. It demonstrates that prior to the GFE implementation, there was no notable difference in the SUGI trends between companies affected by the GFE and those not affected. However, after the GFE was implemented, the SUGI impacted of affected companies improved significantly. These results provide further support for our findings.

**5.3.2. Heterogeneous treatment effects.** Given that GFE policy announcements are not clustered in the same year, varying treatment points over time will subject the traditional two-way fixed effects (TWFE) to potential estimation bias [57–59]. If there is heterogeneity in treatment effects, the TWFE estimator may struggle to accurately identify the true

**Table 3. Baseline regression.**

|  | (1) | (2) |
| --- | --- | --- |
|  | SUGI | SUGI |
| GFE | 0.081*** | 0.076*** |
|  | (0.026) | (0.025) |
| AGE |  | −0.031 |
|  |  | (0.024) |
| SIZE |  | 0.023* |
|  |  | (0.012) |
| MFR |  | −0.044 |
|  |  | (0.043) |
| FLR |  | −0.008 |
|  |  | (0.037) |
| TOP1 |  | −0.000 |
|  |  | (0.001) |
| GO |  | 0.001 |
|  |  | (0.004) |
| PFA |  | 0.147*** |
|  |  | (0.057) |
| PRO |  | −0.169** |
|  |  | (0.067) |
| CFS |  | −0.142*** |
|  |  | (0.053) |
| Firm fixed | Yes | Yes |
| Year fixed | Yes | Yes |
| N | 14446 | 14446 |
| Adj-R² | 0.653 | 0.654 |

Note: ***, **, and * indicate significance at the 1%, 5% and 10% levels, respectively. Values in parentheses are robust standard errors clustered at the firm level. Following tables has the same notes.

average treatment effect. For example, in our study, the total DID estimator over the sample period can be divided into the following three components: (i) New entrants to the GFE (Treatment) and those who have never been selected for the GFE (Never Treated); (ii) New entrants to the GFE (Earlier Group Treatment) and those who have not yet joined the GFE (Later Group Comparison); (iii) New entrants to the GFE (Later Group Treatment) and those who are already participating in the GFE (Earlier Group Comparison). The total DID estimator is derived as a weighted average of the basic 2×2 DID estimates from the three components mentioned above. Among the three control groups, the first two serve as 'good controls' because they assess the impact of whether or not entering the GFE affects firms' SUGI. However, the treatment effect in the third control is not necessarily homogeneous. If the weights assigned to the control mean estimator of the third are larger, this can influence the results of the two-way fixed effects estimator, thereby introducing bias [57,59]. To mitigate the bias problem in TDID estimation caused by heterogeneous treatment effects, we employ the following two methods to analyze heterogeneous treatment effects. To address the bias problem in TDID estimation caused by heterogeneous treatment effects, we employ the following two methods to analyze heterogeneous treatment effects and mitigate the limitations of the TWFE estimator.

First, considering the bias diagnosis of the estimator and Goodman-Bacon decomposition. Following [59], we conducted a bias diagnosis of the TWFE estimator to examine whether the third control group is overrepresented. Fig 6

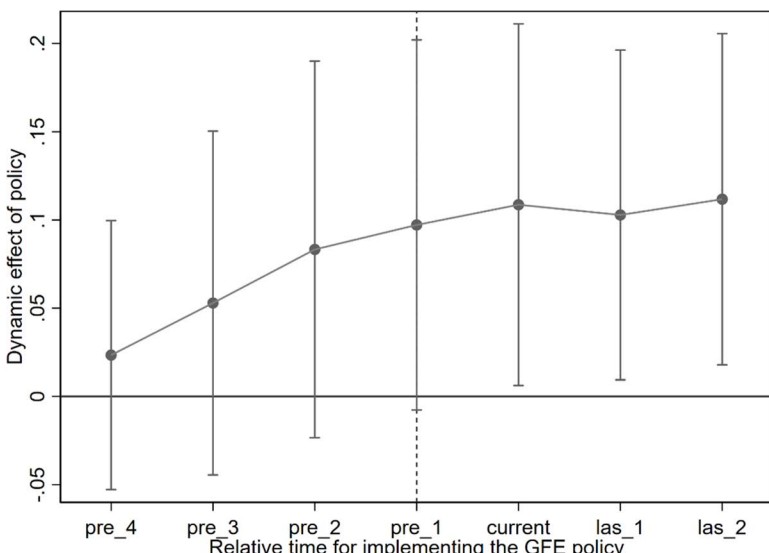

**Fig 5. Parallel trend test.**

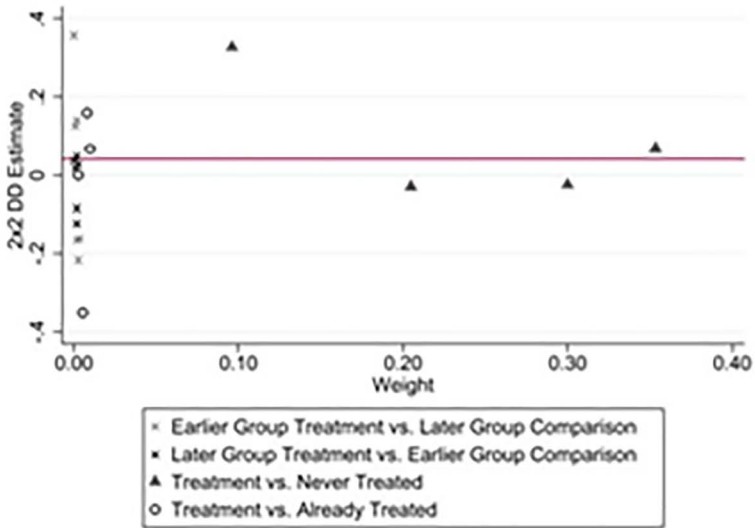

- × Earlier Group Treatment vs. Later Group Comparison
- ▪ Later Group Treatment vs. Earlier Group Comparison
- ▲ Treatment vs. Never Treated
- ○ Treatment vs. Already Treated

**Fig 6. SUGI.**

presents the results of the Goodman-Bacon decomposition, indicating that the third control group constitutes a small proportion in our study. It suggests that the effect of GFE on SUGI is not driven by the third control group. In addition, we provide detailed results of the Goodman-Bacon decomposition in Table 4. It can be observed that the primary source of the overall DID estimates comes from the estimate using the never-treated sample as the control group, which accounts for up to 95% of the weight (Treatment vs. Never Treated). In contrast, the weight of the sample using those who received treatment earlier as the control group is only 0.9% (Later Treatment vs. Earlier Comparison), indicating a minimal impact on the overall estimates. These results suggest that the average treatment effects derived from the benchmark regression results in this study are not significantly biased.

**Table 4. Goodman-Bacon decomposition results.**

| | (1) | (2) |
|---|---|---|
| | SUGI | |
| Control Group Type | Weights | Estimated Value |
| Treatment vs. Never treated | 0.954 | 0.044 |
| Earlier Treatment vs. Later Comparison | 0.011 | −0.084 |
| Later Treatment vs. Earlier Comparison | 0.009 | −0.013 |

Second, considering the heterogeneity-robust estimator. When the heterogeneous treatment effects are more pronounced, the dynamic effects obtained through the classical event study method may also suffer from estimation bias. To address this issue, some econometricians have employed heterogeneity-robust DID estimators to re-estimate the dynamic effects. This approach has been widely adopted in the existing literature [60–61]. Compared to the estimators proposed by [58,62], and others, this estimator exhibits relatively strong estimation properties and is capable of satisfying unbiasedness, validity, and consistency simultaneously. Based on this, we employ the estimation method proposed by [57] to conduct the dynamic effects test. Fig 7 presents the estimation results, demonstrating that the parallel trend assumption remains valid after addressing the bias problem in TDID estimation caused by heterogeneous treatment effects. This confirms that the benchmark regression results in this study are robust and reliable.

**5.3.3. Placebo test.** The promotion effect of GFE on SUGI may be influenced by unobservable factors, resulting in an estimation error. To obtain a more reliable estimate, consistent with [63], we use the placebo test to determine the contingency of GFE's effect. Based on the distribution of GFE in benchmark regression, we randomly sampled 500 times to create a 'pseudo-policy dummy variable' and then re-estimated the relationship between GFE and SUGI using Model (1). The result of the placebo test is shown in Fig 8. These findings bolster the credibility of our estimation results, indicating their high reliability.

**5.3.4. Excluding other policy interference.** In this section, we mitigate the potential influence of extraneous policies on research findings. The low carbon city policy (LCCP), as a comprehensive regulation and policy of government's

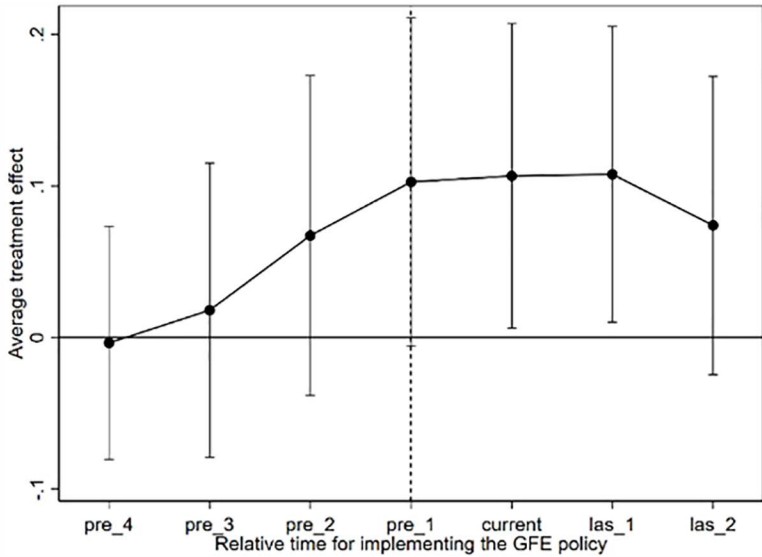

**Fig 7. Parallel trends based on heterogeneity robust DID estimator (Sun and Abraham, 2021).**

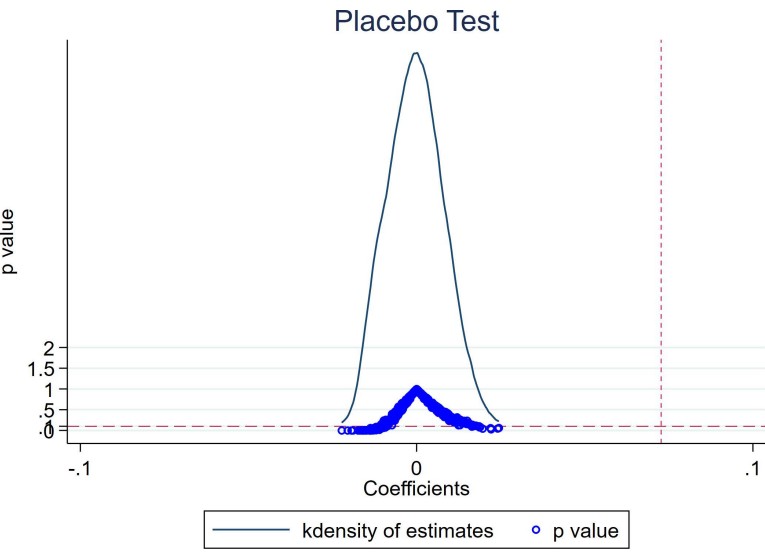

**Fig 8. Placebo test.**

low carbon supervision, aims to restrict the source-of-carbon emission activities of residents and businesses that generate high levels of carbon dioxide. This may potentially influence the green innovation activities of businesses [64]. Simultaneously, taking into account the temporal overlap between LCCP and GFE, we designate LCCP as the dummy variable for the exclusivity test (LCCP = 1, 0 indicates a city hasn't been designated as a low carbon city). The results are presented in Column (1) of Table 5. Upon accounting for the influence of LCCP interference, the regression coefficient of GFE on SUGI remains positive.

Furthermore, the implementation of a stringent environmental protection tax (EPT) may foster the advancement of green technological innovations that apply to the regulation and oversight of environmental pollution. The EPT in China is an essential market-oriented environmental regulation that significantly contributes to the advancement of green transformation. Certain regions have imposed higher taxes on pollutants as a result of the EPT's implementation. Therefore, we consider these regions to be an example of the EPT. Hebei, Jiangsu, Shandong, Henan, Hunan, Sichuan, Chongqing,

**Table 5. Excluding other policy interference.**

|  | (1) | (2) | (3) |
|---|---|---|---|
|  | SUGI | SUGI | SUGI |
| GFE | 0.076*** | 0.076*** | 0.076*** |
|  | (0.025) | (0.025) | (0.025) |
| LCPP | 0.040 |  | 0.040 |
|  | (0.029) |  | (0.029) |
| EPT |  | 0.001 | −0.002 |
|  |  | (0.018) | (0.019) |
| Controls | Yes | Yes | Yes |
| Firm fixed | Yes | Yes | Yes |
| Year fixed | Yes | Yes | Yes |
| N | 14446 | 14446 | 14446 |
| Adj-R² | 0.654 | 0.654 | 0.654 |

Guizhou, Hainan, Guangxi, Shanxi, and Beijing comprise these province regions. Several researchers have discovered that EPT negatively affects green performance [65], greenwashing of businesses [66], and other related factors. To mitigate the potential impact of EPT on the research conclusions, we incorporate EPT as a dummy variable: EPT is assigned a value of 1 when a province or city adopts EPT, and a value of 0 otherwise. Column (2) of Table 5 presents the results. Upon accounting for the influence of EPT interference, the regression coefficient of GFE on SUGI remains statistically significant and positive at the 1% level. Moreover, we also consider the impact of LCCP and EPT. Column (3) of Table 5 shows that when both LCCP and EPT are taken into account simultaneously, the regression coefficient of GFE on SUGI remains notably positive and statistically significant. The results remain consistent with the prior findings.

**5.3.5. Endogeneity analysis.** The enterprises announced by GFE are selected through a series of processes, such as local declaration and expert assessment. As a result, the implementation of GFE may exhibit a degree of randomization across firms. Although we control for firm and time fixed effects as well as a range of control variables, the model may still be affected by firm-time level observable or unobservable omitted variables, which could bias the regression results. To address potential endogeneity issues, we employ the two-stage least squares (2SLS) method. Following [21], we select the interaction term between green space coverage and time in the province where the firm is located as an instrumental variable (IV) for GFE. First, an increase in the proportion of regional green space helps create a favorable ecological environment for the implementation of green factories, ensuring that the IV satisfies the relevance requirement. Second, regional green space is unlikely to directly affect corporate SUGI, allowing the IV to meet the exclusivity requirement. Therefore, the selection of this IV is justified.

Table 6 presents the estimation results of the IV. In Columns (1) and (2), the regression coefficients for 2SLS are 0.024 and 0.151, respectively, both of which are significantly positive at the 1% level. This indicates a strong positive correlation between GFE and the IV selected in this study, suggesting that the IV effectively explain GFE. The results of the weak IV test show that the Kleibergen-Paap rk Wald F statistic exceeds the Stock-Yogo weak IV critical value at the 10% level. Therefore, we reject the weak IV assumption and confirm the validity of this IV. After addressing the endogeneity issue, GFE still significantly promotes SUGI, which supports the core findings of this study.

**Table 6. Endogeneity analysis.**

|  | (1) | (2) |
| --- | --- | --- |
|  | First stage | Second stage |
|  | GFE | SUGI |
| IV | 0.024*** |  |
|  | (0.000) |  |
| GFE |  | 0.151*** |
|  |  | (0.058) |
| Controls | Yes | Yes |
| Firm fixed | Yes | Yes |
| Year fixed | Yes | Yes |
| Industry and region fixed | Yes | Yes |
| N | 15851 | 14637 |
| Adj-$R^2$ | 0.398 | 0.163 |
| Kleibergen-Paap rk LM statistic |  | 3.799 |
| Cragg-Donald Wald F |  | 6442.482 |
| Kleibergen-Paap rk Wald F statistic |  | 30.714 [16.38] |

Note: The values in square brackets represent the critical values at the 10% level for the Stock-Yogo weak identification test.

**5.3.6. Other robustness checks.** We have demonstrated that GFE has a positive effect on SUGI. To enhance the dependability of the research findings, we conduct additional robustness checks by replacing variables, modifying the range of the sample, and adjusting the estimation model.

(1) Replacing the explained variable. To ensure the stability of the SUGI index, we further enrich the measurement method of SUGI. First, according to [67], we calculated the cosine similarity of green invention patent abstracts to reflect SUGI, namely SUGI2. The specific steps for building the cosine similarity index are as follows: First, pre-process the green patent abstract content. This includes removing stop words, punctuation marks, and special characters with no substantive meaning for text analysis and processing to reduce noise interference with index calculation. Next, we use Jieba segmentation to process the text. Divide the text content into multiple words or terms, count their frequency, and create the corresponding word frequency vector. Then, using the 'Term Frequency-Inverse Document Frequency (TF-IDF)' method, assign corresponding weights to each word. This method accurately characterizes the importance of words in text content by combining term frequency (TF) and inverse document frequency. Finally, compute the cosine similarity between the green invention patent abstract content for each company in years t and t-1. The specific calculation formula is as follows:

$$cosine\ similarity_{it} = 1 - \frac{Q_t Q'_{t-1}}{\sqrt{Q_t Q'_{t-1}} \sqrt{Q_t Q'_{t-1}}}, i \neq j$$

(2)

$Q_t$ and $Q_{t-1}$ represent the feature text vectors formed by the green patent abstract text of the company in year $t$ and year $t-1$, respectively. The higher the value, the less similar the contents of patent abstracts in two periods are, implying that the greater the novelty and breakthrough of enterprise green innovation, the more likely it is to break the dependence on technology path, expand the innovation boundary, and reflect the qualitative improvement of green innovation.

Second, drawn on [68–70], we calculated the upper 20% of frequently cited patents and view it as SUGI3. The index is computed using information from the incoPat patent database regarding green invention patents held by publicly traded companies. By utilizing the IPC main classification number information at the sub-group level and the discrepancy in patent classification numbers, we compute the SUGI level of the enterprise by considering the top 20% of green invention patents that were filed in the same application year as high frequency cited patents.

Third, following [71], we calculated the dispersion of green invention patent technology, and define it as SUGI4. The calculation formula is:

$$dispersity_i = 1 - \sum_{j}^{n} S_{ij}^2$$

(3)

$S_{ij}$ represents the proportion of green invention patents applied by company $i$ in patent category $j$, $n$ represents the total number of categories for which company $i$ applied for green invention patents.

Fourth, since patent citations and knowledge breadth can reflect the quality of firms' green innovation [54], following [68,72–75], we measured corporate SUGI through four indicators: (i) The natural logarithm of the cumulative number of citations plus one for green invention patents applied by firms, namely SUGI5. (ii) The natural logarithm of the cumulative citations number of green invention patents applied by firms with self-citation exclusion plus one, namely SUGI6. (iii) The ratio of the number of citations of green invention patents applied by enterprises and that in the industries where the enterprises belong, we define it as SUGI7. (iv) We calculated the knowledge breadth of green invention patents and regard it as SUGI8. The calculation formula is:

$$SUGI8_{it} = 1 - \sum a^2$$

(4)

$SUGI8_{it}$ represents the knowledge breadth of firm $i$'s green invention patents in year $t$. $a$ represents the proportion of each main group in green invention patent classification numbers. A higher value of $SUGI8_{it}$ indicates broader knowledge integration in the green invention patent and consequently higher patent quality. After calculating the knowledge breadth of each green invention patent, we aggregated the measures to the firm level by computing the average values, thereby obtaining the final knowledge breadth of green invention patents.

Fifth, considering the production processes of enterprises may also involve green innovation behaviors, such as improving production processes and management methods, these can also serve as a measure of SUGI. Following [54], we calculated the green management innovation of enterprises and used this indicator as an alternative measure of SUGI, labeled SUGI9. Based on the environmental regulation, certification disclosure table, and management information disclosure table of listed companies in the CSMAR database, we collected data on indicators such as whether a company is ISO14001 certified, whether it is ISO9001 certified, its environmental management system, environmental education and training, and environmental special actions. By aggregating the above indicators, we use the resulting composite score to measure corporate green management innovation.

Sixth, given the limitations of the 2010 WIPO classification in identifying emerging technologies like carbon capture, this classification system may not comprehensively capture all green patents. To address this limitation, we employ the Combined Patent Classification (CPC) system jointly developed by the European Patent Office (EPO) and the United States Patent and Trademark Office (USPTO) to reassess firms' green invention patent. The CPC generally adheres to the classification principles of the IPC while introducing nine divisions (adding division Y to the existing A-H divisions), primarily designed to accommodate emerging technologies and interdisciplinary technologies. Specifically, Y02 is exclusively designated for climate change mitigation and adaptation technologies, serving as a reference standard for identifying green patents. The Y02 classification comprises: Y02A (climate change adaptation technologies), Y02B (building-related energy efficiency technologies), Y02C (carbon capture and sequestration), among others. For detailed information, please refer to: https://www.cooperative-patentclassification.org/cpcSchemeAndDefinitions/table. The CPC classification enables more comprehensive identification of emerging green technologies, thereby compensating for the IPC system's limitations in classifying green patents. Specific steps are as follows: First, we downloaded green invention patent applications with their CPC classification numbers for listed Chinese manufacturing firms from the incoPat patent database. Second, using CPC classification numbers for green technologies, we conducted manual searches and matches in the database to identify green invention patents under the CPC standard. Third, we aggregated annual corporate green patent applications and defined this measure as SUGI10.

Seventh, considering that there may be differences in the applicability of green patents due to the technical characteristics of different industries, the 2010 WIPO classification framework may not precisely reflect industry clusters. For instance, patents containing keywords like 'manufacturing' or 'process' typically belong to the manufacturing patents, while those featuring 'power generation' or 'scheduling' technologies are generally classified as the energy patents. Such categorization bias may lead to the omission of green invention patents in manufacturing sectors, potentially distorting SUGI measurements. To address this limitation, we employ textual analysis methods to cross-validate and supplement the IPC classification results. Specifically, we identify industry-related keywords by screening green invention patent abstracts. When keywords such as 'manufacturing', 'process', or 'production' appear in an abstract, the patent is classified as a manufacturing patent; when terms including 'power grid', 'energy storage', or 'power generation' are detected, it is categorized as an energy patent. Ultimately, we aggregate green invention patents from manufacturing firms identified through both industry keyword screening and IPC classification, using this combined total as a proxy for SUGI, denoted as SUGI11. In Table 7, Columns (1) to (10) present the results of GFE on SUGI2 to SUGI11, which align with the main conclusions.

(2) Changing sample range. Two modifications are made to the sample range in this section. First, the advent of the COVID-19 pandemic has had a profound effect on the economy of China. To mitigate any potential impact on enterprise production behavior, we reassessed the correlation between GFE and SUGI subsequent to the sample exclusion in 2020. The results are presented in Column (1) of Table 8, providing further evidence for the

**Table 7. Other robustness tests.**

| | (1) | (2) | (3) | (4) | (5) | (6) | (7) | (8) | (9) | (10) |
|---|---|---|---|---|---|---|---|---|---|---|
| | SUGI2 | SUGI3 | SUGI4 | SUGI5 | SUGI6 | SUGI7 | SUGI8 | SUGI9 | SUGI10 | SUGI11 |
| GFE | 0.019*** | 0.119*** | 0.116*** | 0.051* | 0.056* | 0.297*** | 0.016** | 0.088** | 0.144*** | 0.070*** |
| | (0.005) | (0.043) | (0.051) | (0.029) | (0.029) | (0.065) | (0.008) | (0.035) | (0.043) | (0.027) |
| Controls | Yes | Yes | Yes | Yes | Yes | Yes | Yes | Yes | Yes | Yes |
| Firm fixed | Yes | Yes | Yes | Yes | Yes | Yes | Yes | Yes | Yes | Yes |
| Year fixed | Yes | Yes | Yes | Yes | Yes | Yes | Yes | Yes | Yes | Yes |
| N | 15671 | 15671 | 15671 | 15671 | 15671 | 13678 | 15671 | 15671 | 15671 | 15671 |
| Adj-R² | 0.555 | 0.609 | 0.439 | 0.764 | 0.741 | 0.380 | 0.478 | 0.603 | 0.637 | 0.710 |

**Table 8. Other robustness tests.**

| | (1) | (2) | (3) | (4) | (5) | (6) | (7) | (8) | (9) |
|---|---|---|---|---|---|---|---|---|---|
| | Changing sample range | | | | | Changing model | | Adding control variables | Other fixed effects |
| GFE | 0.082*** | 0.073*** | 0.105*** | 0.071** | 0.104*** | 0.082*** | 0.073*** | 0.085*** | 0.074*** |
| | (0.028) | (0.026) | (0.032) | (0.031) | (0.031) | (0.028) | (0.026) | (0.029) | (0.024) |
| Controls | Yes | Yes | Yes | Yes | Yes | Yes | Yes | Yes | Yes |
| Firm fixed | Yes | Yes | Yes | Yes | Yes | Yes | Yes | Yes | Yes |
| Year fixed | Yes | Yes | Yes | Yes | Yes | Yes | Yes | Yes | Yes |
| N | 12373 | 13617 | 11809 | 10410 | 12059 | 15671 | 15671 | 12374 | 14446 |
| Adj-R² | 0.645 | 0.654 | 0.657 | 0.672 | 0.657 | 0.555 | 0.439 | 0.660 | 0.657 |

fundamental finding of our research. Second, considering that some regions exhibit unique characteristics in terms of policy implementation, economic development level, industrial structure, and green innovation capacity, we take the following steps to adjust the sample range to eliminate the potential interference of these factors on the research results. (i) Excluding samples from regions with fewer than 10 GFE pilot firms. (ii) To mitigate the influence of regional economic development dynamics, we excluded observations with regional GDP per capita growth rates below the 10th percentile and above the 90th percentile. (iii) Considering that the pollution attributes of firms may influence GFE and thus bias the study's conclusions, we exclude and re-run the regression for samples from regions where highly polluting industries account for more than 30%. (iv) Given that the effects of green innovation require time to materialize and some cities exhibit persistent non-innovation, we excluded cities with zero green patent applications for three consecutive years and re-estimated the regression model. The results of changing sample range are presented in Columns (2) to (5) of Table 8. These results demonstrate that our findings are robust to factors such as regional policy implementation and economic development, further supporting the study's conclusions.

(3) Changing estimation model. There are a large number of 0 in green patent applications. Thus, we select the Tobit and Poisson models to reexamine the impact of GFE on SUGI. The estimation results are presented in Columns (6) and (7) of Table 8. After re-estimating, the estimation coefficients of GFE remain significantly positive.

(4) Adding control variables. Although we control for important variables such as firm age, size, and financial leverage, there may be other unobserved factors that influence both GFE and SUGI, such as a firm's strategic positioning or management practices. Related studies indicate that board and management characteristics significantly affect firms' green innovation activities [76–77]. Furthermore, considering the CEO's environmental background, executives' green perceptions, and the presence of ESG committees may influence corporate governance and strategy, these factors

ultimately affect green innovation [78]. Based on this, we incorporate additional control variables into the model, including board size, the percentage of independent directors, whether directors and supervisors have a financial background, whether they hold dual positions, CEOs' environmental preferences, executives' green perceptions and ESG committee. Following [79–80], we constructed the CEOs' environmental preferences. A CEO's environmental preference is assigned a value of 1 if the CEO possesses green work or educational experience, and 0 otherwise. Specifically, a CEO is considered to have green work experience if their professional background or roles involve environmental protection, energy conservation, or corporate pollution prevention and control. A CEO is considered to have green educational experience if they majored in pulp and paper science, environmental studies, environmental engineering, environmental science, or any other environment-related discipline. The relevant data are sourced from the executive profile module of the CSMAR database. Following [81]'s methodology for measuring executive cognition, we conducted textual analysis of Chinese listed companies' annual reports. We selected relevant keywords for frequency counting and ultimately constructed a measure of corporate executives' green perceptions. The keywords include: energy conservation and emission reduction, environmental strategy, environmental philosophy, environmental management organization, environmental education, environmental training, environmental technology development, environmental audit, energy saving and environmental protection, environmental policy, environmental department, environmental inspection, low-carbon environmental protection, environmental work, environmental governance, environmental protection and governance, environmental facilities, environmental laws and regulations, and pollution control. The ESG committee serves as a key mechanism for the board of directors to address corporate environmental and social matters. Following [82], we calculated a dummy variable for whether the company has an ESG committee. Since the vast majority of companies have not yet established dedicated ESG or CSR committees—instead assigning these responsibilities to committees related to sustainability, environmental protection, or stakeholder rights. Based on this, we expanded our sample selection criteria. We define ESG committee as 1 if a company has any committee whose title or responsibilities include terms such as ESG, CSR, social responsibility, sustainability, green, environmental protection, consumer rights protection or information disclosure, and 0 otherwise. Columns (8) of Table 8 displayed the results, which support the main research conclusions.

(5) Considering other fixed effects. Due to factors such as macroeconomic policies implemented in each region, inter-regional differences in GFE subsidies and the screening of pilot enterprises, as well as economic policies and cyclical changes in the manufacturing industry, there may be heterogeneous impacts on firms' SUGI activities. To address the potential biases caused by these factors, we incorporate region, industry and industry$\times$year fixed effects into the baseline model. Column (9) of Table 8 presents the estimation results, which align with the study's core findings.

### 5.4. Mechanism test

To test the mechanism of how GFE promotes the SUGI of enterprises, this study constructs the following model.

$$M_{it} = \alpha_0 + \alpha_1 GFE_{it} + \alpha_2 X_{it} + \theta_i + \delta_t + \varepsilon_{it} \tag{5}$$

M represents greenwashing and IURO. The definitions of the remaining variables are in line with benchmark regression Model (1).

Greenwashing, being a strategic green behavior, has the potential to impede the green innovation of businesses, particularly the SUGI. In general, when a company divulges environmental information using a qualitative description that lacks substantial operational functionality, it is likely that such disclosure is merely a symbolic approach with considerable ambiguity, potentially driven by a greater intention to engage in greenwashing. Conversely, when a company employs precise figures or other quantitative indicators to characterize pollution discharge, it is generally perceived that such information is of a substantive nature and possesses a high degree of transparency. Thus, the company's incentive to engage in

greenwashing is diminished. On the basis of this information and the manner in which businesses disclose environmental data, we calculate the greenwashing of enterprise. The calculation formula of the index is:

$$greenwashing = \frac{QUAL-QUAN}{The\ number\ of\ items\ that\ have\ been\ disclosed}\#$$

(6)

The number of disclosure items is the total number of environmental liability items that listed companies have voluntarily disclosed. The number of qualitative disclosure items (QUAL) refers to the number of items disclosed in written form. The number of quantitative disclosure items (QUAN) represents the number of items disclosed numerically. Enterprise environmental liabilities include chemical oxygen demand (COD), sulfur dioxide ($SO_2$), carbon dioxide ($CO_2$), smoke dust, and industrial solid waste, among others.

Table 9 shows the regression results. Column (1) shows that the regression coefficient of GFE is −0.116, which is significantly negative at the 1% level, indicating that GFE can improve enterprise SUGI by limiting greenwashing. H2 has been verified. There is no consensus on the relationship between greenwashing and green innovation. Some studies have found that greenwashing negatively affects firms' green innovation, particularly the quality of green innovation [66]. Conversely, other studies suggest that strict environmental regulations may promote green innovation by increasing greenwashing behavior [83]. However, we find that GFE, as a voluntary environmental regulation, can achieve long-term dynamic regulation of firms. By reducing information asymmetry between the government and firms, it mitigates strategic greenwashing behavior and incentivizes firms to engage in higher-quality and SUGI activities.

According to the theoretical analysis, green factory enterprises not only have more resources to engage in open innovation collaboration, but they also send out a lot of green signals to universities and research institutions. This reduces the transaction costs of collaborative innovation between enterprises to some extent. Similar to [84]'s study, we calculate the number of innovation collaborative partners involved in green invention patents to measure IURO. The green innovation collaborative partners include universities, research institutes, and joint laboratories (such as research centers, R&D centers, and technology centers). Column (2) of Table 9 shows the mechanism results of IURO. In Column (2), the estimation coefficient of GFE is 0.023 and significant at the level of 10%, confirming that GFE has notably advanced IURO. The results show that GFE can facilitate corporate SUGI by enhancing IURO, thereby verifying H3. Effective integration of internal and external innovation resources, along with diverse knowledge exchanges across organizational boundaries, is essential to reinforce high levels of innovation within firms [85]. Some studies suggest that IURO enhance the role of intelligent manufacturing in promoting high-quality green innovation in firms [86]. However, the aforementioned study primarily focused on the relationship between intelligent manufacturing pilot policies, IURO, and breakthrough green innovation. Our study further extends their findings. From the perspective of green industrial policy, we find that GFE can also strengthen green technology cooperation between firms and other entities, thereby fully promoting SUGI.

Table 9.  Mechanism analysis.

|  | (1) | (2) |
|---|---|---|
|  | Greenwashing | IURO |
| GFE | −0.116*** | 0.023* |
|  | (0.036) | (0.014) |
| Controls | Yes | Yes |
| Firm fixed | Yes | Yes |
| Year fixed | Yes | Yes |
| N | 15671 | 15671 |
| Adj-R² | 0.318 | 0.311 |

## 5.5. Heterogeneity analysis

### 5.5.1. Can GFE break the path dependence of technology selection in industries with high energy consumption and heavy pollution?.

China's industrial expansion has resulted in a structure with high energy consumption and pollution. Enterprises with high energy consumption and heavy pollution not only play an important role in driving economic development, but they are also critical actors in environmental pollution and governance [87]. However, due to the profit-driven nature of businesses and the non-exclusive nature of environmental governance, these industries generally lack the impetus to drive green innovation. Green innovation, which aims to save resources and energy while also reducing environmental pollution [88], encompasses a variety of innovative activities related to environmental friendliness and sustainable development. SUGI, in particular, can make a significant contribution to green technology by developing and implementing new technologies, materials, and products. It is beneficial for enterprises to significantly increase resource efficiency and achieve green production. However, for traditional manufacturing industries, particularly those with high energy consumption and heavy pollution, which frequently emerged from the framework of the second industrial revolution, the existing inventory of polluting technologies within enterprises may significantly impede the development of clean technologies [89]. Furthermore, a significant amount of pre-fixed capital investment and a number of sunk costs created by these industries may lock in the direction of technological progress [90–91], severely impeding enterprises' green innovation activities. Under these circumstances, how to implement effective green industry policies to encourage SUGI is a topic that both governmental agencies and the academic community must pay close attention to. Can GFE break the path dependence of technology selection in these industries, thereby promoting SUGI? These issues deserve further investigation.

Based on the above analysis, we calculate the proportion of sulfur dioxide emissions in national emissions in 2012 and use it as an indicator of energy consumption in the industry. Following that, we divide the total sample into two categories. Industries with an index higher than the median are classified as HECI, while those with a lower index are classified as LECI. The results are shown in Columns (1) and (2) of Table 10. In Column (1), the coefficient of GFE is significantly positive, while the coefficient of GFE is insignificant at the traditional statistical levels in LECI. It indicates that GFE has a significant influence on HECI's SUGI when compared to LECI. Furthermore, we divide the sample based on pollution intensity into heavy pollution (HPI) and non-heavy pollution (NHPI) industries for comparative analysis. According to the Notice issued by the Ministry of Ecology and Environment of the People's Republic of China (formerly the Ministry of Environmental Protection of the People's Republic of China) regarding the issuance of the 'List of Industry Classifications for Environmental Protection Investigation of Listed Companies', we define the following industries as HPI: The coal mining and washing industry, the petroleum and natural gas extraction industry, the black metal ore mining and dressing industry, the non-ferrous metal ore mining and dressing industry, the textile industry, the leather, fur, feather and its product and footwear industry, the papermaking and paper products industry, the petroleum processing, coking and nuclear fuel

**Table 10. Degree of energy consumption and pollution intensity in industries.**

|  | (1) | (2) | (3) | (4) |
|---|---|---|---|---|
|  | HECI | LECI | HPI | NHPI |
| GFE | 0.071** | 0.057 | 0.094** | 0.071** |
|  | (0.032) | (0.038) | (0.044) | (0.031) |
| Controls | Yes | Yes | Yes | Yes |
| Firm fixed | Yes | Yes | Yes | Yes |
| Year fixed | Yes | Yes | Yes | Yes |
| N | 7924 | 5580 | 4140 | 10280 |
| Adj-R² | 0.610 | 0.689 | 0.574 | 0.674 |

processing industry, the chemical raw materials and chemical products manufacturing industry, the chemical fiber manufacturing industry, the rubber and plastic products industry, the non-metallic mineral products industry, the black metal smelting and pressing industry, the non-ferrous metal smelting and pressing industry, the electricity, heat production and supply industry. In addition, referring to the 'Industry Classification Guidelines for Listed Companies' revised by the China Securities Regulatory Commission (CSRC) in 2012, the industry codes for heavily polluting sectors in our research within the manufacturing industry are C17, C19, C22, C25, C26, C28, C29, C30, C31, and C32, respectively. Others are NHPI.

The estimation results are shown in Table 10, Columns (3) and (4). In Column (3), the results show that the GFE coefficient is positive and significant at the 5% level. It indicates that, when compared to NHPI, GFE has a significant impact on the SUGI of enterprises in HPI. To summarize, GFE will break the path dependence of technology selection in these industries, thereby significantly increasing the SUGI of enterprise in HECI and HPI. Our study enriches the research on the green innovation effects of environmental policies in the heavy chemical industries. Bi [92] demonstrate that although the implementation of environmental protection laws strengthens firms' green innovation, it does not enhance SUGI. However, our study reveals that GFE can break the path dependence of technological choices in high energy consumption and heavy pollution industries, significantly enhancing corporate SUGI.

We believe that there are several plausible explanations for these results. First, because of their unique characteristics of high energy consumption and heavy pollution, these industries are generally under a lot of pressure to carry out green innovation. However, the implementation of GFE will allow enterprises to receive more support from government policy while also sending positive green signals to the market. This process will assist enterprises in obtaining additional financial support, optimizing and upgrading production equipment, thereby promoting SUGI. Second, unlike other industries, industries with high pollution and energy consumption have a higher green innovation cost, and there is obvious greenwashing, which may seriously impede enterprises' green innovation activities. Nonetheless, GFE can effectively inhibit greenwashing by enterprises, then significantly promoting SUGI in high pollution and high energy consumption industries.

**5.5.2. How does the disclosure of external information affect GFE's relationship with SUGI?.** There is a significant information asymmetry between enterprises and governments. As a result, enterprises can use green gimmicks to carry out low-quality innovation activities while receiving government subsidies [93]. Although GFE emphasizes whole-chain monitoring as well as fairness and transparency in assessment, which can significantly reduce enterprise strategic behavior, it cannot completely address information asymmetry. Some studies suggest that better external information disclosure can reduce information asymmetry and prevent opportunistic behaviors among enterprises [94]. External information disclosure by enterprises may create a favorable environment for GFE to play a more effective role, resulting in a significant positive impact on GFE's green effect. For example, media attention can create favorable conditions for GFE's enterprise to play a more effective role.

According to some studies, enterprises that receive a lot of media attention have better external information disclosure and are more likely to attract investors [95]. As a result of media attention, external enterprise supervision has become more effective. Enterprises are motivated to carry out SUGI activities in response to high media attention and a desire to improve their social reputation. Moreover, a number of studies have found that institutional investors have a significant impact on enterprises' green innovation capabilities [96–97]. Furth, some studies show that analysts have distinct information advantages and greater information transparency when tracking enterprises [98–99]. These findings can be attributed to the fact that analysts can obtain more information through visits, surveys, and other channels, allowing investors to better monitor the company's green innovation activities [100]. In these circumstances, whether the influence of GFE on SUGI is 'aimless' or 'the right medicine' depends on enterprise information disclosure.

Based on the previous analysis, our study investigates enterprises' external information disclosure from three perspectives: media attention, institutional investment, and analyst attention. To begin, we calculate the number of times an enterprise is reported by online media as a measure of information transparency [101]. Based on the median of media reports, we divide the entire sample into high media attention (HMA) and low media attention (LMA) groups. The

estimation results are shown in Columns (1) and (2) of Table 11. It indicates that GFE plays a significant role in promoting the SUGI of enterprises in the HMA sample, but not in the LMA sample at the traditional statistical level. Second, as a significant shareholder in the company, institutional investors can participate in internal decision-making and have more complete information, which is beneficial to the company's external disclosure. Based on the median shareholding ratio of institutional investors, we divide the total sample into two categories: high institutional shareholding (HIS) and low institutional shareholding (LIS). The results for the aforementioned categories are shown in Columns (3) and (4) of Table 11. In Column HIS, GFE's regression coefficient on SUGI is positive and obvious, but its influence on SUGI is insignificant in Column LIS. Finally, we run grouping tests from the perspective of analysts. Because analysts benefit from information and technology, the greater the number of analysts, the less information asymmetry, and the higher the quality of external information disclosure. As a result, we calculate the number of analysts who follow the enterprise as a proxy for analyst attention. We divide the entire sample into two groups based on the median: high analyst attention (HAA) and low analyst attention (LAA). The regression results in Columns (5) and (6) of Table 11 show that GFE has a significant impact on the SUGI of enterprises in HAA, but no effect in LAA. These findings suggest that, GFE can significantly improve SUGI for enterprises with good external information disclosure. As a critical component of the environmental governance system, environmental information disclosure serves as a key driver for promoting corporate green innovation [102]. Some scholars have highlighted the significant role of high-level disclosure in the process of green credit policies fostering corporate green innovation [103]. Based on this, our study finds that stronger disclosure can also synergize with GFE to jointly enhance corporate SUGI.

## 6. Conclusions and policy implications

### 6.1. Conclusion

SUGI refers to a company's long-term investment in green technology research and development, aimed at achieving breakthroughs that genuinely reduce pollutant emissions and enhance environmental benefits, thereby keeping a long-term competitive advantage for the company [104]. As a core initiative to promote the green manufacturing system, GFE provides an effective approach to enhance SUGI. We treat GFE as a quasi-natural experiment and empirically examine the relationship between GFE and SUGI by using the TDID method. The main findings are summarized as follows:

First, GFE plays a significant role in promoting SUGI in manufacturing firms. Liu [21] demonstrated that GFE can enhance green innovation, providing valuable insights for our study. This study focuses on substantive technological innovation behavior within green innovation [7,22,105]. By integrating GFE and SUGI into the same research framework, we find that GFE significantly enhances firms' SUGI. This not only enriches the discussion on the environmental performance effects of GFE but also provides new insights and directions for deepening theoretical research in the field of green innovation.

Table 11. External information disclosure of enterprises.

| | (1) | (2) | (3) | (4) | (5) | (6) |
|---|---|---|---|---|---|---|
| | HMA | LMA | HIS | LIS | HAA | LAA |
| GFE | 0.138*** | 0.005 | 0.100*** | 0.037 | 0.082** | 0.004 |
| | (0.042) | (0.024) | (0.036) | (0.035) | (0.032) | (0.043) |
| Controls | Yes | Yes | Yes | Yes | Yes | Yes |
| Firm fixed | Yes | Yes | Yes | Yes | Yes | Yes |
| Year fixed | Yes | Yes | Yes | Yes | Yes | Yes |
| N | 6887 | 6942 | 6989 | 7170 | 9888 | 3898 |
| Adj-R² | 0.681 | 0.620 | 0.693 | 0.601 | 0.678 | 0.608 |

Second, after confirming the positive impact of GFE on firms' SUGI, we further explore the mechanisms through which GFE affects SUGI. Unlike existing studies that examine the impact mechanisms of green industrial policies on SUGI in terms of environmental investment and human capital investment [22], this study focuses on GFE and identifies the critical roles of greenwashing and IURO in promoting SUGI. Through these mechanisms, GFE addresses and mitigates issues such as strategic greenwashing and insufficient resources for green R&D faced by firms during production, significantly enhancing SUGI. These findings enrich research on the mechanisms through which GFE promotes SUGI and provide new insights and directions for exploring the channels through which macro-pilot policies influence corporate green innovation.

Third, GFE has a stronger facilitating effect on SUGI in high energy consumption and heavy pollution industries, effectively breaking the path dependence of technological choices in these sectors. David [90] argues that path dependence in technological innovation means that prior technological innovations shape subsequent technological trajectories. This implies that in heavy chemical industries, firms' existing stock of polluting technologies may hinder the adoption of cleaner technologies [89], thereby obstructing green development. However, few existing studies have explored this issue from the perspective of green industrial policy. We find that GFE significantly contributes to SUGI in high energy consumption and heavy pollution industries, effectively breaking the path dependence of technological choices in these sectors. Additionally, we demonstrate that GFE significantly enhances SUGI among firms with stronger external information disclosure. This suggests that GFE can synergize with external information disclosure to promote SUGI. These findings provide new empirical evidence for researchers to analyze the role of GFE in influencing SUGI across different contexts.

## 6.2. Practical implications

Our findings provide valuable policy suggestions for promoting green industrial policy and facilitating the green transformation of manufacturing in emerging economies.

First, the government should adopt proactive green industrial policies to strengthen corporate SUGI by promoting GFE. Our research demonstrates that GFE can significantly enhance SUGI. On the one hand, governments should develop a forward-looking, multi-level, and systematic policy framework to enhance the top-level design of green industrial policies. This will help integrate industrial policies with environmental development and provide an effective policy environment for enterprises to achieve high-quality green innovation and development. On the other hand, the government should also summarize the successful experiences of GFE, further refine and expand the scope of green factory evaluation, and fully leverage the nationwide driving effect of GFE. These measures will help provide sufficient motivational support to advance SUGI in the manufacturing industry.

Second, the government should effectively streamline the channels for implementing GFE to promote SUGI. Mechanism analysis indicates that GFE can enhance corporate SUGI by mitigating greenwashing and strengthening IURO. The government should continue to enhance the environmental regulatory system. By strengthening laws, regulations, and institutional norms, it can improve the level of environmental information disclosure, thereby reducing enterprises' short-term strategic behaviors such as greenwashing and pseudo-green practices. In addition, the government should actively promote IURO among green factory enterprises and encourage universities, research institutions, and enterprises to establish broader green partnerships. For example, the government can encourage the establishment of industry-university-research collaboration laboratories, green technology research institutes, and green innovation platforms involving green factory enterprises, universities, and research institutions. The creation of a green cooperative ecosystem will provide pilot enterprises with access to advanced green technologies and information exchange, thereby offering an effective pathway for green factory enterprises to promote SUGI.

Third, the government should focus on fully considering the heterogeneous characteristics of enterprises and encourage heavy chemical industries to actively participate in GFE to strengthen the impact of GFE on enhancing SUGI. Our empirical results demonstrate that GFE significantly enhances corporate SUGI in heavily polluting and energy-intensive

industries. Specifically, the government should formulate targeted measures to promote the green transformation of heavy chemical industries, focusing on production processes, production flows, and other aspects. This will accelerate the green transformation of the entire production chain by increasing the stock of clean technologies. For example, the government should strictly enforce GFE standards for heavy chemical industries and strengthen the supervision and management of production processes for enterprises that have been certified as green factories. This will help these industries break the path dependence on traditional technological choices, thereby enhancing SUGI.

### 6.3. Limitation and future research directions

Our study has some limitations that suggest directions for future research. First, we use green invention patents to measure SUGI. Although we have also constructed a number of indicators to test the robustness of SUGI from different perspectives, such as the cosine similarity of green patents and the number of citations, there is not yet a broad consensus in the academic community on these measures. As the economics of innovation evolves, we will need to further explore the construction of SUGI in the future to more accurately and comprehensively reflect firms' SUGI. Second, our study focuses on listed manufacturing companies in emerging developing countries, which may limit the generalizability of the findings. In future research, we plan to collect samples from firms in more countries and regions for comparative analysis. This will enable us to provide more comprehensive empirical evidence to identify similarities and differences in the relationship between green industrial policies and SUGI.

### Acknowledgments

The authors would like to express their gratitude to all peer reviewers for their reviews and comments.

### Author contributions

**Conceptualization:** Xiangshu Dong, Yongjiao Du.

**Investigation:** Xiangshu Dong, Xiang Xiao.

**Methodology:** Yongjiao Du.

**Resources:** Xiang Xiao.

**Software:** Xiang Xiao.

**Supervision:** Xiangshu Dong.

**Validation:** Yongjiao Du.

**Writing – original draft:** Yongjiao Du.

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
