## [Decision Letter · Decision Letter 0]

26 Feb 2025

How does China’s green factory policy affect substantive green innovation?

PLOS ONE

Dear Dr. Xiao,

Thank you for submitting your manuscript to PLOS ONE. In view of the referees’ feedback and my own reading of your paper, we invite you to address all issues noted below. Although the reviewers consider that the manuscript requires a minor revision, we believe that these issues are major in nature, requiring more than a superficial or minor revision.

We have particular concerns about the underlaying hypothesis, the robustness of the methods and analysis and the soundness and basis of the conclusions.

Since our point of view the paper has an important potential to be consider for publication on this journal, so we invite you to address the issues noted below and resubmit the manuscript for a new revision round.

Comments from the editorial office: We note that one or more reviewers has recommended that you cite specific previously published works. As always, we recommend that you please review and evaluate the requested works to determine whether they are relevant and should be cited. It is not a requirement to cite these works. We appreciate your attention to this request.''

We look forward to receiving your revised manuscript.

Kind regards,

Juan E. Trinidad-Segovia, PhD

Section Editor

PLOS ONE

Journal Requirements:

2. Please include captions for your Supporting Information files at the end of your manuscript, and update any in-text citations to match accordingly. Please see our Supporting Information guidelines for more information: http://journals.plos.org/plosone/s/supporting-information .

3. We are unable to open your Supporting Information file data.dta and do.do. Please kindly revise as necessary and re-upload.

Reviewers' comments:

Reviewer's Responses to Questions

**Comments to the Author**

1. Is the manuscript technically sound, and do the data support the conclusions?

Reviewer #1: Partly

Reviewer #2: Yes

2. Has the statistical analysis been performed appropriately and rigorously?

Reviewer #1: Yes

Reviewer #2: Yes

3. Have the authors made all data underlying the findings in their manuscript fully available?

Reviewer #1: Yes

Reviewer #2: Yes

4. Is the manuscript presented in an intelligible fashion and written in standard English?

Reviewer #1: Yes

Reviewer #2: Yes

Reviewer #1: This paper investigates the impact of China’s Green Factory Evaluation (GFE) policy on firms' substantive green innovation (SUGI). The findings show that GFE can significantly enhance corporate SUGI. The proposed mechanism suggests that GFE promotes corporate SUGI by limiting greenwashing and fostering industry-university-research collaboration. Additionally, the paper highlights that GFE plays a significant role in promoting SUGI in HECI and HPI, as well as in firms with high levels of information disclosure.

My comments are as follows:

1) There are widespread concerns about biases in staggered difference-in-differences regression estimators. As mentioned in the paper, “from 2017 to 2020, the MIIT identified five batches of green factory lists and evaluated them at different times, as stated in the General Office of the MIIT’s Notice on Developing Green Manufacturing System and related documents.” The paper covers the period from 2012 to 2021 and employs a staggered difference-in-differences strategy. However, when treatment effects change over time, the standard staggered DD regressions can introduce a “bad comparisons” problem, leading to significant estimates with the wrong sign. The econometric literature has proposed several alternative DD estimation techniques, such as Callaway and Sant'Anna (2021) and Sun and Abraham (2021), to circumvent the problems with TWFE staggered DD estimators. And there are applied studies using these estimators as a remedy. Therefore, I suggest at least one of these estimators be used in this paper (at least in the section of robustness checks) to provide more valid estimates.

2) The criteria for defining green invention patents and their representativeness are questionable. First, the authors do not provide a clear explanation of the standards used to define green invention patents. Given that different classification systems or evaluation frameworks could lead to inconsistencies in the number of green patents filed by firms, this lack of clarity raises concerns about the validity of the measure. Second, it is important to note that not all green invention patents necessarily represent substantial technological innovation. Some patents may involve only minor technical improvements, which may not have significant practical market value. While invention patents are generally considered to be of higher quality, variations in patent quality across firms could lead to biased or inaccurate estimates. Moreover, firms' green innovation activities may include other forms of innovation, such as the adoption of green technologies or improvements in production processes, which are not adequately captured by patent filings alone. I recommend that the authors clarify the criteria used to define green invention patents and consider using more comprehensive and representative indicators to measure firms' actual green innovation.

3) The baseline regression model (Model 1) exhibits endogeneity issues. First, omitted variable bias may contribute to endogeneity. Although the paper controls for several key variables (e.g., firm age, size, and financial leverage), there may be other unobserved factors that simultaneously influence both the Green Factory Evaluation (GFE) policy and substantive green innovation (SUGI), such as a firm’s strategic positioning or management practices. Second, endogeneity resulting from the policy selection process could also be problematic. The implementation of the GFE policy (i.e., assignment to the treatment group) may not be entirely random, as it could be influenced by unobservable firm-specific characteristics. Firms may decide to participate in the green factory certification process based on their own conditions (such as resources, technological capabilities, or prior levels of green innovation), and these characteristics are likely correlated with their green innovation outcomes. This could lead to selection bias, where better-performing firms are more likely to be designated as green factories. Third, the omission of industry and regional fixed effects may introduce additional endogeneity issues. Variations in regional subsidies for green factories and differences in policy effects across industries could distort the estimated results. To address these concerns, I recommend using instrumental variables to mitigate endogeneity and incorporating industry and regional fixed effects to improve the robustness and accuracy of the model.

4�The authors' approach of excluding samples based solely on urban administrative structure in the robustness checks lacks methodological rigor. Instead, the authors should consider excluding regions that exhibit distinctive characteristics, such as variations in policy implementation, economic development levels, industrial structures, or green innovation capabilities, rather than arbitrarily removing samples based on administrative structure. This would ensure that the exclusions are more relevant and grounded in factors that could truly affect the results of the study.

Reviewer #2: The manuscript (PONE-D-24-52239) entitled “How does China’s green factory policy affect substantive green innovation?” provides an empirical analysis on studying the nexus between China's Green factory policy and green innovation using the firm-level panel data collected between 2012 and 2021. The authors mainly utilized staggered difference-in-differences to establish the causality. The main results show that green factory evaluation can significantly improve corporate substantive green innovation. Overall, the manuscript is original and interesting. I have several minor suggestions for the authors (see the attachments).

**Do you want your identity to be public for this peer review?** For information about this choice, including consent withdrawal, please see our Privacy Policy

Reviewer #1: No

Reviewer #2: No

---

## [Author Response · Author response to Decision Letter 1]

23 Mar 2025

Dear Editor,

Thank you for your patience and guidance! As suggested by you, we have re-uploaded the fully anonymized dataset to the public database Figshare and the DOI link is: https://doi.org/10.6084/m9.figshare.28646624. Have a nice day!

Dear Reviewer 1,

Thank you very much for giving us the opportunity to further improve the quality of our manuscript entitled “How does China’s green factory policy affect substantive green innovation?” (PONE-D-24-52239). We deeply appreciate your valuable concerns and suggestions, which enable us to further improve our paper. We have read through all comments from you and tried our best to revise the entire paper as suggested. The responses to your comments are displayed in the “Response to Reviewers”, and the corresponding revisions are shown in “Revised Manuscript with Track Changes”. Have a nice day!

Dear Reviewer 2,

Thank you very much for giving us the opportunity to further improve the quality of our manuscript entitled “How does China’s green factory policy affect substantive green innovation?” (PONE-D-24-52239). We deeply appreciate your valuable concerns and suggestions, which enable us to further improve our paper. We have read through all comments from you and tried our best to revise the entire paper as suggested. The responses to your comments are displayed in the “Response to Reviewers”, and the corresponding revisions are shown in “Revised Manuscript with Track Changes”. Have a nice day!

---

## [Decision Letter · Decision Letter 1]

15 Apr 2025

PONE-D-24-52239R1How does China’s green factory policy affect substantive green innovation?PLOS ONE?

Dear Dr. Xiao,

We look forward to receiving your revised manuscript.

Kind regards,

Juan E. Trinidad-Segovia, PhD

Section Editor

PLOS ONE

Journal Requirements:

Reviewers' comments:

Reviewer's Responses to Questions

**Comments to the Author**

Reviewer #1: All comments have been addressed

Reviewer #2: All comments have been addressed

2. Is the manuscript technically sound, and do the data support the conclusions?

Reviewer #1: Partly

Reviewer #2: Yes

3. Has the statistical analysis been performed appropriately and rigorously?

Reviewer #1: Yes

Reviewer #2: Yes

4. Have the authors made all data underlying the findings in their manuscript fully available?

Reviewer #1: Yes

Reviewer #2: Yes

5. Is the manuscript presented in an intelligible fashion and written in standard English?

Reviewer #1: Yes

Reviewer #2: Yes

Reviewer #1: Your revisions significantly enhanced the quality of the study. To further improve the manuscript, the following suggestions are offered for your consideration:

1. On the definition and representativeness of green invention patents: you have made a reasonable effort to distinguish green invention patents from utility models by referencing China’s Patent Law and the WIPO Green Classification. However, I would suggest further clarifying certain methodological choices. For instance, your reliance on the 2010 WIPO classification might introduce temporal limitations, as emerging technologies like carbon capture (prominent post-2020) could be underrepresented. Could you elaborate on the rationale for selecting the 2010 version or address its potential constraints in covering newer innovations? Additionally, while applying the WIPO framework, your analysis does not explicitly discuss whether industry-specific adjustments or sensitivity tests were conducted, particularly given sectoral clustering (e.g., energy vs. manufacturing patents). A brief discussion on the classification’s applicability across industries would strengthen your methodology.

2. Addressing omitted variable bias: regarding robustness checks, your use of authorized invention patent counts might conflate high-impact innovations with incremental improvements. Could you consider incorporating quality indicators like patent citations or claims to better differentiate these? Furthermore, while addressing omitted variable bias, including industry×year fixed effects might more effectively control for time-varying sectoral shocks (e.g., post-2015 environmental inspections). Your current model also omits firm-level factors like CEO environmental preferences or ESG committees—exploring these in supplementary analyses could enhance robustness.

3. Refining sample exclusion criteria: your exclusion criteria based on cross-sectional thresholds (e.g., single-year GDP percentiles) might overlook persistent confounders. Would expanding these to a dynamic framework (e.g., excluding cities with no green patents for three consecutive years) better address such concerns? Addressing these points would further clarify your study’s rigor and generalizability.

Reviewer #2: The authors have well adressed all my concerns in revised manuscript and I donot have further comments. I believe it now meets the requirements for publication in the PLOS ONE journal.

**Do you want your identity to be public for this peer review?** For information about this choice, including consent withdrawal, please see our Privacy Policy

Reviewer #1: No

Reviewer #2: No

---

## [Author Response · Author response to Decision Letter 2]

24 Apr 2025

Dear Editor,

Thank you very much for giving us the opportunity to further improve the quality of our manuscript entitled “How does China’s green factory policy affect substantive green innovation?” (PONE-D-24-52239R1). We deeply appreciate your valuable concerns and suggestions, which enable us to further improve our paper. We have read through all comments from you and tried our best to revise the entire paper as suggested. Revised portions are marked with red words in “Revised Manuscript with Track Changes”. Once again, we would like to express our gratitude to you for your patience and guidance. Have a nice day!

Dear Reviewer 1,

Thank you very much for giving us the opportunity to further improve the quality of our manuscript entitled “How does China’s green factory policy affect substantive green innovation?” (PONE-D-24-52239R1). We deeply appreciate your valuable concerns and suggestions, which enable us to further improve our paper. We have read through all comments from you and tried our best to revise the entire paper as suggested. Revised portions are marked with red words in “Revised Manuscript with Track Changes”. Have a nice day!

Dear Reviewer 2,

Thank you very much for your careful and professional review on “How does China’s green factory policy affect substantive green innovation?” (PONE-D-24-52239R1). Your comments and suggestions were invaluable and they greatly improved the quality of the paper. Your endorsement helps us tremendously! Once again, we would like to express our gratitude to you for your patience and guidance. Have a nice day!

---

## [Decision Letter · Decision Letter 2]

14 May 2025

How does China’s green factory policy affect substantive green innovation?

PONE-D-24-52239R2

Dear Dr. Xiao,

We’re pleased to inform you that your manuscript has been judged scientifically suitable for publication and will be formally accepted for publication once it meets all outstanding technical requirements.

Kind regards,

Juan E. Trinidad-Segovia, PhD

Section Editor

PLOS ONE

Additional Editor Comments (optional):

Reviewers' comments:

Reviewer's Responses to Questions

**Comments to the Author**

Reviewer #1: All comments have been addressed

2. Is the manuscript technically sound, and do the data support the conclusions?

Reviewer #1: Yes

3. Has the statistical analysis been performed appropriately and rigorously?

Reviewer #1: Yes

4. Have the authors made all data underlying the findings in their manuscript fully available?

Reviewer #1: Yes

5. Is the manuscript presented in an intelligible fashion and written in standard English?

Reviewer #1: Yes

Reviewer #1: The authors have thoroughly addressed my comments. They have also included additional robustness checks and alternative measures of SUGI to enhance the study's reliability. The revisions have significantly improved the manuscript's quality and addressed the concerns raised. I appreciate their efforts and the improvements made.

**Do you want your identity to be public for this peer review?** For information about this choice, including consent withdrawal, please see our Privacy Policy

Reviewer #1: No

---

## [Editor Report · Acceptance letter]

PONE-D-24-52239R2

PLOS ONE

Dear Dr. Xiao,

I'm pleased to inform you that your manuscript has been deemed suitable for publication in PLOS ONE. Congratulations! Your manuscript is now being handed over to our production team.

Kind regards,

on behalf of

Dr. Juan E. Trinidad-Segovia

Section Editor

PLOS ONE